# Overcoming the Fibrotic Fortress in Pancreatic Ductal Adenocarcinoma: Challenges and Opportunities

**DOI:** 10.3390/cancers15082354

**Published:** 2023-04-18

**Authors:** Kay K. Myo Min, Charlie B. Ffrench, Claire F. Jessup, Mia Shepherdson, Savio George Barreto, Claudine S. Bonder

**Affiliations:** 1Centre for Cancer Biology, University of South Australia and SA Pathology, Adelaide, SA 5000, Australia; kay.myomin@unisa.edu.au (K.K.M.M.); charlie.ffrench@mymail.unisa.edu.au (C.B.F.); 2College of Medicine & Public Health, Flinders University, Bedford Park, SA 5042, Australia; 3Adelaide Medical School, University of Adelaide, Adelaide, SA 5000, Australia; 4Hepatopancreatobiliary & Liver Transplant Unit, Division of Surgery & Perioperative Medicine, Flinders Medical Centre, Bedford Park, SA 5042, Australia

**Keywords:** pancreatic ductal adenocarcinoma, tumour microenvironment, desmoplasia, immunotherapy, targeted therapy

## Abstract

**Simple Summary:**

Pancreatic ductal adenocarcinoma (PDAC) is the most common form of pancreatic tumours, representing greater than 90% of all diagnosed cases of pancreatic cancer. PDAC has the lowest 5-year survival rate of all tumour malignancies with less than 9% patient survival. A unique feature of PDAC tumours is the presence of a dense fibrotic fortress that creates a physical barrier around the cancer cells, thus resulting in a reduced penetrability of drugs and a ‘sanctuary’ in which cancer cells thrive, termed desmoplasia. Extensive desmoplasia in the PDAC tumour microenvironment (TME) is a crucial factor that influences PDAC development, progression, metastasis, and resistance to treatment. This review will focus on the role of the TME in PDAC, current treatments for PDAC, and a reflection on past and current clinical trials targeting components of the TME in PDAC.

**Abstract:**

An overabundance of desmoplasia in the tumour microenvironment (TME) is one of the defining features that influences pancreatic ductal adenocarcinoma (PDAC) development, progression, metastasis, and treatment resistance. Desmoplasia is characterised by the recruitment and activation of fibroblasts, heightened extracellular matrix deposition (ECM) and reduced blood supply, as well as increased inflammation through an influx of inflammatory cells and cytokines, creating an intrinsically immunosuppressive TME with low immunogenic potential. Herein, we review the development of PDAC, the drivers that initiate and/or sustain the progression of the disease and the complex and interwoven nature of the cellular and acellular components that come together to make PDAC one of the most aggressive and difficult to treat cancers. We review the challenges in delivering drugs into the fortress of PDAC tumours in concentrations that are therapeutic due to the presence of a highly fibrotic and immunosuppressive TME. Taken together, we present further support for continued/renewed efforts focusing on aspects of the extremely dense and complex TME of PDAC to improve the efficacy of therapy for better patient outcomes.

## 1. Introduction

Pancreatic ductal adenocarcinoma (PDAC) is the most common form of pancreatic neoplasms, representing greater than 90% of all diagnosed cases of pancreatic cancer with neuroendocrine tumours accounting for the remaining 10% [1]. PDAC has the lowest 5-year survival rate of all tumour malignancies with less than 9% patient survival [2]. Poor survival of patients with PDAC is largely attributed to the cancer developing with few symptoms and as a result, most PDAC patients are diagnosed with an advanced stage of disease, often including metastases to the liver, lungs, and peritoneum [3,4,5]. Unfortunately, the incidence of PDAC is on the rise. In the United States, the incidence of pancreatic cancer is projected to reach the second leading cause of cancer-related death (behind lung cancer) by 2040 and will surpass breast, prostate, and colorectal cancers [2,6].

## 2. Disease Development

PDAC originates from the exocrine portion of the pancreas which constitutes 90% of the organ and consists of acinar cells and a ductal network. Acinar cells exhibit a degree of plasticity which allows them to contribute to both homeostasis and regeneration of pancreatic exocrine tissue. Under environmental stimuli (e.g., tissue damage, inflammation, stress) acinar cells transdifferentiate to a more epithelial phenotype, a process known as acinar to ductal metaplasia (ADM) [7,8]. During ADM, acinar cells are more susceptible to mutational hits, which can accumulate and lead to the development of precursor neoplasms and invasive adenocarcinoma (Figure 1). Pancreatic intraepithelial neoplasms (PanINs) are the most common form of pancreatic precursor lesions. The three grades of PanINs, namely, PanIN-1A (flat) and PanIN-1B (papillary), PanIN-2 and PanIN-3 consist of increasing amounts of cell atypia [9]. Additionally, intraductal papillary mucinous neoplasms (IPMNs) and mucinous cystic neoplasms (MCNs) are cystic pancreatic lesions that can also lead to invasive PDAC [7,9]. The accumulation of genetic changes (in several oncogenic and tumour suppressor genes) within these cells leads to an invasive PDAC [7].

## 3. Drivers

A unique feature of PDAC tumours is an abundance of dense fibrotic stroma that constitutes up to 90% of the tumour volume [10,11]. The dense desmoplastic stroma creates a physical barrier around the cancer cells, fortifying the epithelium and hindering vascularisation, thus resulting in reduced penetrability of drugs and a ‘sanctuary’ in which cancer cells thrive [12]. Extensive desmoplasia in the tumour microenvironment (TME) is a crucial factor that influences PDAC development, progression, metastasis, and resistance to treatment [13] and is arguably the most important pathophysiological feature of this cancer.

PDAC is documented to have a complex genomic landscape and mutation profile of cancer associated genes [14,15]. During tumour development, key oncogenic alterations occur where the genes *KRAS* (Kirsten rat sarcoma viral oncogene homolog), *CDKN2A* (cyclin dependent kinase inhibitor 2A), *SMAD4* (Mothers against decapentaplegic homolog 4), and *TP53* (Tumour protein 53) are most frequently mutated [4,16].

Key signalling pathways have been implicated in the development of PDAC and the development of desmoplasia. In particular, the tumour growth factor (TGF)-β signalling pathway plays an important role in pancreatic carcinogenesis [17]. TGF-β also regulates the development of the desmoplastic TME by inducing a fibroblast response which causes dense build-up of stromal cells and extracellular matrix (ECM) [18]. Cancer associated fibroblasts (CAFs) are the primary producer of the ECM protein collagen which is one of the main contributors of tumour desmoplasia in PDAC [19,20].

Integrin signalling is also an important cornerstone of PDAC TME homeostasis as it mediates adhesion of cells to ECM components such as collagen (types I and IV), fibronectin, and laminin [21]. As the main receptors for ECM molecules, integrins engage with the actin cytoskeleton via the linker protein talin and initiate bidirectional signals from both sides of the plasma membrane. Integrins can function collectively as mechanoreceptors in response to extracellular mechanical signals such as sheer stress and transduce the signal into the cell (outside-in signalling) [22]. Integrins are also known to regulate cancer cell activation and recruitment of CAFs in the stroma through TGF-β and participate in ECM remodelling (via the platelet derived growth factor receptor (PDGFR)), to provide a pre-metastatic niche for invading cancer cells (inside-out signalling) [23,24]. The bi-directional signalling of integrins is critical for promoting and regulating cell proliferation, adhesion, and migration, as well as chemotherapy resistance [21,25].

Signalling through the Janus kinase (JAK)/signal transducer and activator of transcription (STAT) pathway is initiated upon binding of cytokines such as interleukin (IL)-1 and IL-6 to their receptor, and is reported to enhance PDAC fibrosis, and ECM remodelling for increased stiffness of PDAC tumours [26,27,28]. In PDAC, IL-1α-induced signalling leads to activation of JAK1/2 and STAT1/3 in ECM producing pancreatic stellate cells which then promotes an inflammatory state of CAFs favouring further tumour growth [29].

Sonic hedgehog (SHH) is also overexpressed by neoplastic PDAC cells, and this pathway is important for the formation of a fibroblast-rich desmoplastic stroma [30]. The increased expression of SHH has been observed during pancreatic tumourigenesis, where the expression becomes elevated as the tumour progresses towards an advanced stage of PDAC, and aberrant activation of the Hedgehog signalling pathway is reported to be associated with metastatic potential of the tumour [3,27,31].

## 4. Tumour Microenvironment

The TME of PDAC tumours is characterised by the recruitment and activation of CAFs, heightened ECM deposition (e.g., collagen), reduced blood supply, as well as increased inflammation via the influx of inflammatory cells and cytokines into the tumour (Figure 2) [32].

### 4.1. Fibroblasts

The PDAC TME consists of an abundance of stromal cells such as pancreatic stellate cells and CAFs, which are major contributors to the dense desmoplastic reaction seen in PDAC through the production of acellular components that make up the ECM [33,34]. Stellate cells are myofibroblast-like cells that produce large amounts of ECM components, growth factors, and cytokines/chemokines in support of PDAC growth [35,36,37]. Bachem, Schünemann [38] and colleagues documented that when stellate cells were grown in the conditioned media of PDAC cancer cell lines, there was an increase in cell proliferation and synthesis of collagen type I and fibronectin. When the opposite occurred (i.e., PDAC cells were grown in the conditioned media of stellate cells), PDAC cells demosnstrated increased proliferation, invasion and migration [39]. Furthermore, mice injected with CAFs and PDAC cells produced larger tumours with increased desmoplasia and fibrosis when compared to PDAC cells injected alone [13,38,40].

CAFs are major orchestrators of the pancreatic TME due to their dominant production of acellular components of the ECM, e.g., collagen [33,34]. In the stroma, CAFs interact with other cells as well as the ECM to mediate cancer cell proliferation, migration, invasion, and metastasis, in part through reorganisation of the ECM. Within the PDAC tumour, there exists a tumour-stroma crosstalk between cancer cells and CAFs through paracrine signalling which influences tumour progression and desmoplasia through the production of growth factors, cytokines, and chemokines [32]. For instance, factors such as IL-6 and IL-1 (both of which induce JAK/STAT signalling) are secreted by cancer cells to promote CAF activation [29,41]. In response, CAFs then secrete chemokines, cytokines (e.g., IL-6), growth factors (e.g., vascular endothelial growth factor, VEGF), exosomes, and metabolites to instruct cancer cells and other TME components that promote angiogenesis and recruit immunosuppressive cells for immune cell evasion; thus further promoting PDAC progression [32,42,43]. The release of the growth factor TGF-β by cancer cells also induces the formation of a desmoplastic ECM produced by CAFs, and up-regulation of integrins on CAFs. Integrin α3β1 binds laminin-332 to mediate CAF activation and promote PDAC growth and invasion. Integrins also assist in the assembly of fibronectin at CAF cell protrusions to further enable integrin-mediated cancer cell migration [34,44,45]. Taken together, there exists a complex relationship between PDAC cells, CAFs, stellate cells, the TME, the immune cell response and cancer progression.

### 4.2. Extracellular Matrix

The ECM is a non-cellular component of the TME which provides essential physical scaffolding as well as biochemical and biomechanical cues to the cells [46]. ECM components include collagen, fibronectin, proteoglycans, and hyaluronic acid, as well as catalytically active enzymes and proteinases [47]. The connective tissue of PDAC can make up 60–90% of the total tumour area [48], and its presence correlates with poor survival for PDAC patients [49].

In PDAC progression and in pancreatitis, collagens are the most prominent component of ECM proteins, comprising more than 90% of the ECM matrisome at all stages of disease. Proteoglycans and glyco-proteins such as fibronectin make up the rest of the matrix and these ECM proteins are all overexpressed in PDAC [20]. An analysis of tumour extracts also reveal that of the 28 different collagen subtypes (categorised into four subfamilies), PDAC tumours contain an overabundance of fibril-forming collagen subfamily types I, III, and V, and network-forming collagen subfamily type IV [50,51]. High collagen content in PDAC is reported to correlate with reduced patient survival [52]. Collagens I, IV, and V promote the pro-tumourigenic features of PDAC cell survival, proliferation, migration, invasion and EMT [53,54,55]. In PDAC tumours, collagen I is the most abundant ECM protein in the TME and has a high affinity for signalling via integrin β1 that is expressed on the surface of PDAC cells. More specifically, integrins β1 and β6 are reported to promote PDAC cell growth, survival, migration, and invasion [46,56,57]. Type IV collagen exerts its pro-tumorigenic effects on the pancreatic cancer cells via an autocrine loop, where it interacts with integrin receptors on the surface of the cancer cells to enhance their survival [53]. Similarly, type V collagen influences cancer cells in a paracrine fashion via activation of the β1-integrin/focal adhesion kinase (FAK) signalling pathway [58]. Indicating that the cellular and acellular components in the TME work in tandem to maintain and regulate cancer cell growth.

Tumour stiffness caused by collagen cross-linking can also enhance cancer cell proliferation and promote cancer cell invasion via integrin signalling. Briefly, collagen cross-linking occurs via a family of enzymes, lysyl oxidases (LOX), which activate signalling factors such as yes-associated protein (YAP) and transcriptional coactivator with PDZ-binding motif (TAZ) for cancer cell proliferation [59,60]. An increase in collagen thickness and tissue tension is also linked to PDAC prognosis [28,61]. Notably, increased collagen content, accompanied by an increase in another ECM protein hyaluronan (HA), both contribute to drug resistance (e.g., doxorubicin) in pancreatic cancer [62]. Notably, the desmoplastic collagen in PDAC is not all tumour promoting but can also be a barrier to cancer development. By way of example, a study by Rhim, Oberstein [30] showed that reducing stromal desmoplasia in the TME of PDAC tumours (by inhibiting the SHH pathway component Smoothened) accelerated tumour growth and metastasis.

### 4.3. Vasculature

As tumours increase in size, the TME becomes increasingly hypoxic, and cancer cells adapt by upregulating tissue processes that enhance access to the blood supply [4,63]. One such process, and another hallmark of cancer progression, is the induction of neo-angiogenesis (the formation of new blood vessels from existing endothelial cell lined vessels). Angiogenesis is a process largely driven by the signalling protein VEGF [64] which is upregulated in response to hypoxia [7,65].

Interestingly, some tumours, including PDAC, are considered ‘hypo-vascular’ and are populated with avascular stromal ‘deserts’ with substantially lower microvessel densities when compared to healthy areas of the pancreas [66]. Vascular dysfunction presents as a major obstacle to pharmaco-delivery and drug efficacy in the fight against PDAC; Komar, Kauhanen [67] and colleagues documented that compared to adjacent normal pancreatic tissue, the blood flow in pancreatic tumours is decreased by ~60% which consequently infers poor blood perfusion and impaired drug delivery to the tumour site [67].

Despite the dysfunctional vasculature and poor blood perfusion, PDAC tumours are still able to thrive by adapting and using alternate metabolic and scavenging pathways (e.g., autophagy and micropinocytosis) [68,69]. To this end, studies have shown that some solid tumours, including PDAC, are able to gain access to the blood circulatory system via a method independent of endothelial cell lined vasculature, by forming their own vessel-like structures in a process known as vasculogenic mimicry (VM) [70,71]. Studies have shown that a VM phenotype has been associated with poor prognosis and that PDAC cancer cells have the ability to form functional VM vessels in vitro and in vivo [71,72]. While much is still to be uncovered about the VM potential of PDAC cells, Yang, Zhu [73] demonstrated that hypoxia-inducible factor 2 alpha (HIF-2α) promotes VM formation in vitro and in vivo through twist family basic helix-loop-helix (bHLH) transcription 1 (Twist1) binding to vascular endothelial-cadherin (VE-cadherin) in pancreatic cancer cells. Our own work has also highlighted the expression of adhesion molecules by VM-competent cancer cells allowing for the active recruitment of circulating leukocyte subsets [74].

### 4.4. Immune Cell Presence

PDAC tumours are often considered immunologically ‘cold’; however, the role of the immune TME in PDAC is emerging as an important prognostic feature [75]. The PDAC TME is characterised by a highly heterogeneous immune cell infiltration profile: in the early stages of PDAC development, the TME is largely pro-inflammatory, but following the infiltration of immunosuppressive cells, the TME shifts to an anti-inflammatory state [76]. When this occurs, the PDAC TME has a high content of regulatory T cells (Tregs), tumour associated macrophages (TAMs), and myeloid-derived suppressor cells (MDSCs), as well as a relatively low prevalence of anti-tumour CD4+ and CD8+ T cells, natural kill (NK cells), and dendritic cells (DCs) [34].

#### 4.4.1. Conventional T Lymphocytes

T lymphocytes are central to tumour immunology, being able to specifically recognise tumour-associated antigens. Circulating effector, memory and effector memory T cells are all able to migrate from the blood to tumour sites. Conventional cytotoxic CD8+ T cells are powerful effectors of the antitumour immune response, capable of directly killing cancer cells by secreting granules containing enzymes including granzymes and perforin. The infiltration of tumours by CD8+ T cells is associated with an improved prognosis in multiple cancer types including PDAC [77,78]. While many studies have focussed on CD8+ T cells, the importance of CD4+ T cells for tumour control and response to immunotherapy approaches is becoming apparent [79]. CD4+ helper T cells license antigen-presenting cells (APC) to engage in effective priming of CD8 T cells, as well as activating NK cells, myeloid cells, and other cell types via secreted factors. There is growing evidence for a cytotoxic subset of CD4+ T cells capable of directly killing cancer cells. Conventional subsets of CD4+ T cells including T helper type 1 (T_H_1), T_H_2, T_H_17, T_H_9, T follicular helper (T_FH_) and Tregs are found within tumours and numbers of CD4+ T cells with T_H_1 phenotype are associated with beneficial outcomes. In addition to recruited populations of T cells, tissue resident memory T cells (T_RM_) are a tissue-specific component of the TME. T_RM_ are CD4 or CD8 positive T lymphocytes that persist in tissues long-term following primary T cell responses, and their proportion within the T cell infiltrate is related to a good prognosis [80]. A recent PDAC study of tumour infiltrating lymphocytes identified significant populations of CD8+ T_RM_ with an exhausted (PD1 ^high^, TIGIT ^high^) phenotype, Tregs and T_H_17 cells in line with an immunosuppressed microenvironment [81]. Furthermore, it has been shown that both the infiltration of CD4+ T cells and CD8+ T cells is associated with improved overall survival (OS) and disease-free survival (DFS) in PDAC; however, the TME of PDAC tumours have poor infiltration of CD4+ and CD8+ lymphocytes [82,83].

#### 4.4.2. Regulatory T Lymphocytes

Tregs are a subpopulation of CD3 + CD4 + CD25+ T cells that express the transcription factor forkhead box P3 (FOXP3) and have a role in maintaining homeostasis and self-tolerance. In PDAC, Tregs are considered pro-tumorigenic due to their ability to promote the TME development, promote cancer cell invasion and facilitate anti-tumour immune escape [84]. The number of tumour infiltrating Tregs gradually increases throughout the progression of PDAC and is strongly associated with poor prognosis due to their ability to suppress tumour specific CD4+ and CD8+ T cells and NK cells [85]. Tregs have been documented to directly modulate the TME through the production of IL-10, TGF-β, IL-35 and granzyme B, and indirectly through the expression of cytotoxic lymphocyte-associated antigen-4 (CTLA-4) and programmed cell death 1 (PD-1) [84]. IL-10 and IL-35 are immunosuppressive cytokines produced by Tregs that inhibit effector immune cells (i.e., CD4+ and CD8+ T cells), therefore protecting the cancer cells from immune surveillance [84,86]. Tregs can also modulate immunosuppression through the CTLA-4 and PD-1 pathways. CTLA-4 can bind at higher affinity to CD80 and CD86 ligands expressed on APCs than CD28 which is the co-factor essential for activation of naïve T cells. Bengsch, Knoblock [87] demonstrated that targeting CD25, using an anti-CD25 antibody in PDAC, reduced the number of Treg cells within the tumour and increased the number of CD4+ T helper cells, reaffirming that depleting Tregs revitalises the immune system for an anti-tumour immune response.

#### 4.4.3. B Lymphocytes

B cells are the key component of the adaptive humoral immune system and function by producing antigen-specific antibodies against non-self [76]. Relevant to tumour immunology, they are also able to present antigen to T cells within the TME [88]. The role of B cells within the PDAC TME remains to be fully elucidated, with some studies suggesting a pro-tumourigenic role and others suggesting an anti-tumourigenic role. For example, an in vivo study by Spear, Candido [89] observed that B cells have an immunosuppressive role when in secondary lymphoid organs, however, are more immunostimulatory when found in the PDAC TME and therefore support the anti-tumour immune response [89]. Castino, Cortese [90] also demonstrated that within PDAC tumours B cells are found in two histologically different structures, either as infiltrating lymphocytes or in organised tertiary lymphoid tissue (TLTs). An increase in OS for PDAC patients was only associated with B cells found in TLTs, and this was correlated with a higher number of infiltrating CD8+ T cells; however, infiltrating B cells within the TME was associated with a poorer OS [90]. Clearly, there is a need for further studies to better elucidate the function of B cells in PDAC development.

#### 4.4.4. Natural Killer Cells

NK cells are innate cytolytic immune cells that recognise and directly kill cancer cells through receptor activation. NK cells are also able to elicit an anti-tumour response through interactions with other immune cell types (e.g., DCs, macrophages and T cells) [91]. There are two main subtypes of NK cells: CD56^dim^ which exert potent cytotoxicity and secrete low levels of cytokines (e.g., interferon gamma (IFNγ), tumour necrosis factor alpha (TNFα)), and CD56 ^bright^ which are poorly cytotoxic and secrete high levels of cytokines [85]. In PDAC, NK cells are characterised by impaired anti-tumour activity as well as reduced expression of cytotoxicity receptors [91]. Peng, Zhang [92] demonstrated that the percentage of surface receptors and cytotoxic granules (e.g., perforin and granzyme B) were significantly downregulated in NK cells following exposure to PDAC cancer cells. It is thought that the impaired NK activity is due to secreted factors such as TGF-β and IL-10, as well as indoleamine 2,3-dioxygenase (IDO) and matrix metalloproteinases (MMPs) [91,92]. Furthermore, a study by Lim, Kim [93] and colleagues demonstrated that NK cells isolated from PDAC patient tumours had reduced expression of C-X-C motif chemokine receptor 2 (CXCR-2) when compared to healthy donor NKs, which is suggested to be responsible for the low numbers of NK cells seen within the PDAC TME [93].

#### 4.4.5. Tumour Associated Macrophages

TAMs are a type of myeloid cell that are found in high proportions within the PDAC TME [94]. TAMs are known to exert both pro-tumourigenic and anti-tumourigenic roles depending on the polarised phenotype of the macrophage [95]. In the early stages of PDAC the TME is characterised by macrophages that have a M1 phenotype (i.e., pro-inflammatory) [95,96]. M1 TAMs are attracted to the TME by cytokines such as IFNγ and TNFα where they secrete proinflammatory cytokines, chemokines and effector molecules to intensify the tumouricidal activity [97]. As PDAC disease progresses, TAMs are influenced by the release of cytokines such as IL-4, IL-10, IL-13 and TGF-β and as a result polarise to a M2 phenotype (i.e., anti-inflammatory) [98]. The M2 phenotype is documented to promote immunosuppression, angiogenesis, ECM remodelling and desmoplasia, accelerate metastasis, and overall progression of disease [99]. M2 macrophages produce IL-10, TGF-β, chemokine (C-C motif) ligand (CCL) 2, CCL17 and other cytokines/chemokines that inhibit the activity of CD8+ cytotoxic T cells and NK cells whilst promoting the migration of Tregs into the tumour [97]. Interestingly, Liou, Döppler [100] showed that the depletion of macrophages or the neutralisation of macrophage attracting intercellular adhesion molecule 1 (ICAM-1) delayed the development of PanIN lesions, therefore revealing a role for TAMs within the PDAC TME and a potential therapeutic avenue.

#### 4.4.6. Myeloid-Derived Suppressor Cells

MDSCs are a heterogenous population of activated myeloid progenitor cells that are formed in the process of myelopoiesis within the bone marrow [101]. During homeostasis, immature myeloid cells (IMCs) differentiate into lineage-specific cell populations. However, in a cancer setting, the overproduction of soluble factors such as granulocyte macrophage colony-stimulating factor (GM-CSF), granulocyte colony-stimulating factor (G-CSF), IL-6, VEGF and TNFα promotes the formation of MDSCs that are recruited to the tumour site via the CXC family of chemokines such as CCL2, CXCL-12, CXCL-15 [102]. MDSCs can be characterised into two distinct populations; monocytic MDSCs (M-MDSCs) and granulocytic MDSCs (G-MDSCs) [96]. MDSCs are known to contribute to the immunosuppressive TME within PDAC through the direct and indirect inhibition of T cells and NK cells, and the cross-talk and stimulation of Treg cells [91,96]. It has also been demonstrated that compared to healthy controls PDAC patients have elevated levels of MDSCs within the bone marrow and blood circulation which are rapidly recruited to the PDAC TME [103]. Furthermore, it has been shown that targeting G-MDSCs in PDAC increased the numbers of activated cytotoxic T cells within the PDAC TME, whilst also inducing apoptosis of cancer cells and remodelling of the TME (particularly the stroma) [104].

#### 4.4.7. Dendritic Cells

DCs are professional APCs that form a critical link between the innate and adaptive immune systems [85]. Tumour-specific immunity is mediated by DCs as they recognise, process, and then present tumour associated antigens to adaptive cells [76]. DCs are rarely found within the PDAC TME, but rather in the stroma surrounding the tumour. DCs in the TME are often dysfunctional due to cancer cell secreted factors (e.g., IL-6, VEGF, TGF-β, and reactive oxygen species (ROSs)) [105]. These dysfunctional DCs are compromised in their ability to engulf, process, and present tumour-specific antigens, thereby inhibiting an anti-tumour response [85]. When DCs are present in high numbers within the peripheral circulation and the PDAC TME, there is improved OS of the patients [106]. Furthermore, Fukunaga, Miyamoto [107] demonstrated that the number of DCs within the PDAC TME was significantly increased when CD4+ and CD8+ TILs were also present.

## 5. Current Treatments and Hurdles

Currently, PDAC treatment outcomes are determined by the disease stage at presentation. The treatment associated with the best OS is a margin-negative surgical resection including chemo- and/or radiotherapy. However, this is only achieved in ~10–20% of patients who are diagnosed at a relatively early stage. The remaining 80–90% of patients present with advanced, non-resectable disease and a majority possessing distant metastasis [7,108]. Advanced PDAC tumours are currently treated with chemotherapeutics such as gemcitabine, capecitabine and 5-fluorouracil [109]. FOLFIRINOX (oxaliplatin, irinotecan, leucovorin and fluorouracil) or a combination therapy of gemcitabine and a nanoparticle albumin-bound paclitaxel (nab-paclitaxel) have been employed and have demonstrated an improved survival in a higher proportion of patients when compared to gemcitabine alone [110]. Unfortunately, these treatments are associated with increased toxicity and hence, can only be prescribed in patients with a good performance index [111]. In addition to the late diagnosis of PDAC patients, there are also intrinsic and extrinsic factors that can impede upon therapy success. Such examples include acquired chemotherapy resistance, high mutational burden and pro-oncogenic signalling potential by the cancer cells, high metastatic burden, and the presence of an immunosuppressive TME. Furthermore, the dense stromal and connective tissue in PDAC tumours contributes to elevated tissue pressure through eliciting solid stress by compressing on the blood vessels, further inhibiting effective penetration of anti-cancer drugs, and thus contributing to a lack of efficiency for PDAC treatments [12,112,113,114]. Thus, the OS for PDAC patients has not improved in decades highlighting the need for more potent therapies and earlier detection methods.

### Clinical Trials

Erlotinib (epidermal growth factor receptor (EGFR) tyrosine kinase inhibitor), and Olaparib (a poly adenosine diphosphate (ADP)-ribose polymerase (PARP) inhibitor) are the only two approved targeted therapies used for the treatment of PDAC (Table 1, Figure 2) [115]. However, most patients have, or inevitably develop, intrinsic resistance to EGFR inhibitors, leading to disease progression [116]. In fact, as a survival mechanism, tumours have reportedly developed resistance through the activation of EGFR-independent signalling pathways, which are downstream of the erythroblastic leukaemia viral oncogene homologue (ErbB) family members that promote persistent cancer cell survival [116]. Olarparib is also approved for PDAC patients with breast cancer gene 1 *(BRCA1)* or *BRCA2* germline mutations in their cancer cells, which reportedly only reflects a small subset of patients diagnosed with metastatic pancreatic cancer [117,118].

Historically, clinical trials with targeted treatments for PDAC have been underwhelming. Multiple clinical trials of anti-angiogenic agents that target the tumour vasculature without interfering with the normal vasculature yielded disappointing results (Table 2, Figure 2) [112,119]. Even anti-cancer therapies proven to be successful in the treatment of other cancers such as Trametinib, a MEK (mitogen-activated protein kinase kinase) inhibitor were found to be ineffective in PDAC [27]. Additionally, given that the hedgehog signalling pathway is key in PDAC development, SHH inhibition was also tested, but without success. Further investigation in murine studies revealed that complete inhibition of SHH led to an increase in vascularity and proliferation, showing that Hedgehog driven tumours also supress tumour growth by restraining tumour angiogenesis [30]. Due to the immunosuppressive nature of the PDAC TME, several clinical trials have investigated immune checkpoint inhibitors (ICIs), anti-PD-1 or anti-programmed death ligand-1 (PD-L1) (that block the inhibition of T cell activation by cancer cells) as both a monotherapy and in combination with other therapies, i.e., gemcitabine); however, results have thus far been disappointing (Table 2) [120,121,122,123,124].

The dense stromal reaction leading to the compression of blood vessels and inadequate drug penetration has also emerged as a potential target for therapeutic intervention. A preclinical study demonstrated that excessive HA present in PDAC tumours (which elevates interstitial pressure and impairs perfusion) is degraded by pegvorhyaluronidase alfa (PEGPH20), and that this degradation of HA by PEGPH20 led to an increase in drug delivery. Another clinical trial targeted ECM producing pancreatic stellate cells via all-trans-retinoic acid (ATRA) to reprogram pancreatic stroma and suppress PDAC growth. ATRA in combination with gemcitabine-nab-paclitaxel chemotherapy delivered to patients with advanced, unresectable PDAC is currently in a phase II randomised trial [125]. Another approach utilised Losartan, a clinically approved angiotensin II receptor antagonist with antifibrotic activity (i.e., reduce the amount of collagen and HA in the tumour) [126], which led to decompression of tumour vessels and significantly improved perfusion [127]. Losartan has also been reported to enhance immune activation, and currently a randomized phase II study on combining chemoradiotherapy and losartan with ICI immunotherapy nivolumab (anti-PD-1 blocking antibody) is in progress [128]. A clinical trial is also underway testing the efficacy of targeting connective tissue growth factor (CTGF). In preclinical mouse models, treatment with pamrevlumab (or FG-3019), a humanized monoclonal antibody targeting CTGF, shows that pamrevlumab attenuates tumour growth, metastasis, and angiogenesis [129]. The clinical trial testing pamrevlumab in combination with gemcitabine-nab-paclitaxel or FOLFIRINOX in patients with locally advanced PDAC is currently ongoing.

Given that PDAC tumours are intrinsically immunosuppressive and have lower immunogenic potential [34], targeting the tumour promoting immune cells is an attractive approach to inhibit PDAC progression. To this end, current clinical trials are investigating novel immune-modulating agents such as those targeting CAF-mediated immunosuppression, checkpoint inhibitors, myeloid cells, Tregs, stromal depletion (by targeting CAF, FAK, PDGFRα, IL-1, IL-6), and chimeric antigen receptor (CAR)-T cells (e.g., carcinoembryonic antigen (CEA), mesothelin (MSLN) and mucin 1 (MUC1)) (Figure 2). These agents are being considered as a monotherapy as well as in combination with standard of care chemotherapeutic drugs [130,131]. Immunotherapy aims to improve the anti-tumour immune response and has proven successful in other solid cancers (e.g., melanoma); however, this has not been the case in PDAC [132]. Several immunotherapies of different categories including immunomodulators (e.g., ICIs), immune stimulatory agonists, cytokines and adjuvants), oncolytic viruses, monoclonal antibodies (mAbs), adoptive cell therapies and cancer vaccines have and are being assessed in PDAC [132].

**Table 1 cancers-15-02354-t001:** Non-chemotherapy targeted drug clinical trials that progressed to approval.

Trial [Reference]	Year	Target	Trial Design	Comparator Groups	Overall Survival	Inference
NCT00026338 (NCIC CTG PA.3) [115]	2001–2004	HER1/EGFR	Phase III Randomised Triple blinded Parallel assignment	Erlotinib and gem (*n* = 285) vs. Placebo and gem (*n* = 284)	OS was significantly longer in the erlotinib and gem group with an estimated HR of 0.82 (95% CI, 0.69 to 0.99; *p* = 0.038) PFS was significantly longer in the erlotinib and gem group with an estimated HR of 0.77 (95% CI, 0.64 to 0.92; *p* = 0.004)	Erlotinib improves OS and PFS when used concurrently with gem
NCT02184195 [118]	2014–2019	PARP	Phase III Randomised Quadruple blinded Parallel assignment	Olaparib (*n* = 92) vs. Placebo (*n* = 62)	No significant difference in OS PFS was significantly longer in the olaparib group than in the placebo group HR of 0.53 (95% CI, 0.35 to 0.82; *p* = 0.004)	Olaparib improves PFS to patients with germline *BRCA* mutated PDAC

Abbreviations: CI–confidence interval, HER1/EGFR–human epidermal growth factor receptor type 1, HR–hazard ratio, OS–overall survival, PFS–progression free survival.

**Table 2 cancers-15-02354-t002:** Failed clinical trials targeting the PDAC tumour microenvironment.

Trial [Reference]	Year	Target	Trial Design	Comparator Groups	Overall Survival	Inference
Anti-angiogenic agents						
NCT00088894 (CALGB 80303) [133]	2004–2006	VEGFA	Phase III Randomised Double blinded Parallel assignment	Bevacizumab and gem (*n* = 302) vs. Placebo and gem (*n* = 300)	No significant difference in OS and PFS	The combination of bevacizumab and gem does not improve survival in PDAC
NCT0095966 (BAY 43-9006) [134]	2004–2006	Raf serine/threonine kinase isoforms, VEGFR2, receptor tyrosine kinases	Phase III Not blinded Single group assignment	Sorafenib and gem (*n* = 17)	Trial terminated–lack of efficacy	The combination of sorafenib and gem does not improve survival in PDAC
VMIN 000005133 (PEGASUS-PC study) [135]	2009–2014	VEGFR2	Phase II/III Randomised Double blinded Parallel assignment	Elpamotide and gem (*n* = 100) vs. Placebo and gem (*n* = 53)	No significant difference in OS No significant difference in PFS	The combination of elpamotide and gem does not improve survival in PDAC
Anti-fibrotic/ECM agents						
NCT01231581 [136]	2010–2012	MEK1/2	Phase II Randomised Double-blinded Parallel assignment	Trametinib and gem (*n* = 80) vs. Placebo and gem (*n* = 80)	No significant difference in OS and PFS	The combination of trametinib and gem does not improve survival in PDAC
NCT01130142 [137]	2010–2012	Hedgehog pathway	Phase I/II Randomised Double-blinded Parallel assignment	Saridegib (IPI-926) and gem vs. Placebo and gem Total *n* = 122	Trial terminated–patient survival diminished (increase in vascularisation and proliferation)	The combination of saridegib and gem diminishes patient survival
NCT01064622 [138]	2009–2012	Hedgehog pathway (smoothened)	Phase II Randomised Double-blinded Parallel assignment	Vismodegib and gem (*n* = 53) vs. Placebo and gem (*n* = 53)	No significant difference in OS and PFS	The combination of vismodegib and gem does not improve survival in PDAC
NCT02715804 (HALO-301) [139]	2016–2019	Hyaluronan	Phase III Randomised Double-blinded Parallel assignment	PEGPH20 and AG (*n* = 327) vs. Placebo and AG (*n* = 165)	No significant difference in OS and PFS	The combination of PEGPH20 and AG does not improve survival in PDAC
NCT01821729 [140]	2013–2018	Renin-angiotensin system	Phase II Not blinded Single group assignment	FOLFIRINOX, losartan and proton beam RT (*n* = 49)	69% of participants became eligible for tumour resection	FOLFIRINOX, losartan and proton beam RT was well tolerated, further studies required
Immunological agents						
NCT02558894 [121]	2015–2017	PD-L1 CTLA-4	Phase II Randomised Not blinded Parallel assignment	Durvalumab alone (*n* = 33) vs. Durvalumab and tremelimumab (*n* = 32)	No significant difference in OS and PFS	The combination of durvalumab and tremelimumab does not improve survival in PDAC
Brahmer, 2013 NCT00729664 [122]	2009–2015	PD-L1	Phase I Non-randomised Not blinded Parallel assignment	MDX1105-01 at increasing concentrations (*n* = 14 PDAC patients)	No objective response to treatment	MDX1105-01 does not improve survival in PDAC
NCT00112580 [141]	2005–2009	CTLA-4	Phase II Not blinded Single group assignment	Ipilimumab (*n* = 27)	No or minor objective response to treatment	Ipilimumab does not improve survival in PDAC
NCT00084383 [142]	2002–2005	GM-CSF	Phase II Not blinded Single group assignment	GVAX and adjuvant chemoradiotherapy (*n* = 60)	Median OS of 24.8 months (95% CI, 21.2–31.6)	GVAX and adjuvant chemotherapy was well tolerated, further studies required
NCT02562898 [143]	2015–2019	B cells (reprogramming)	Phase I/II Non-randomised Not blinded Parallel assignment	Ibrutinib, paclitaxel and gem (*n* = 18)	No objective response to treatment	Ibrutinib, paclitaxel and gem does not improve survival in PDAC

Abbreviations: CI–confidence interval, FOLFIRINOX–oxaliplatin, irinotecan, leucovorin and fluorouracil, Gem–Gemcitabine, HER1/EGFR–human epidermal growth factor receptor type 1, HR–hazard ratio, MEK1/2–Mitogen-activated protein kinase 1/2, OS–overall survival, PFS–progression free survival, RAF–rapidly accelerated fibrosarcoma.

## 6. Conclusions

While the survival rates of most cancers have dramatically improved in the last few decades, this is not the case for PDAC where the 5-year survival rate has remained below 9% [144]. At the present time, achieving an improved survival rate for patients with PDAC is fraught with barriers including the lack of a specific biomarker to detect PDAC at an early stage (when meaningful attempts at treatment are feasible), and the inability to identify patients who will benefit from aggressive treatments (surgery and chemo-/radio-therapy) [145]. Unfortunately, even the most exciting advances in cancer treatments have been disappointing in PDAC. For example, while immunotherapy (e.g., ICIs such as anti-PD-1 antibodies) has transformed the treatment landscape for many cancer patients (e.g., lung, melanoma), for PDAC patients, <2% show improved outcomes.

As detailed above, the main challenge is the difficulty in delivering drugs to the tumour, with the presence of a densely packed ‘desmoplastic’ TME preventing most anti-PDAC drugs and immune cells from reaching the ‘heart’ of the tumour in therapeutic/effective concentration. Given that the prognosis of PDAC patients is largely determined by stage of diagnosis and tumour biology [146], renewed efforts focusing on aspects of the extremely dense and complex TME provide hope for improved efficacy of PDAC therapy.

## Figures and Tables

**Figure 1 cancers-15-02354-f001:**
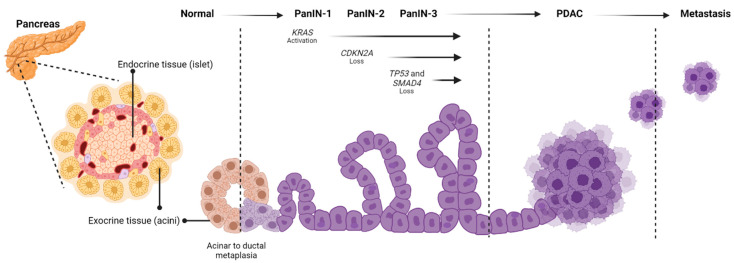
Progression and development of PDAC. Acinar cells in a normal pancreatic duct transform into ductal epithelial cells during the development of pancreatic intraepithelial neoplasia (PanIN) stages 1–3, which can further develop into pancreatic ductal adenocarcinoma (PDAC) after key oncogenic genetic mutation events (KRAS activation, CDKN2A/TP53/SMAD4 deactivation).

**Figure 2 cancers-15-02354-f002:**
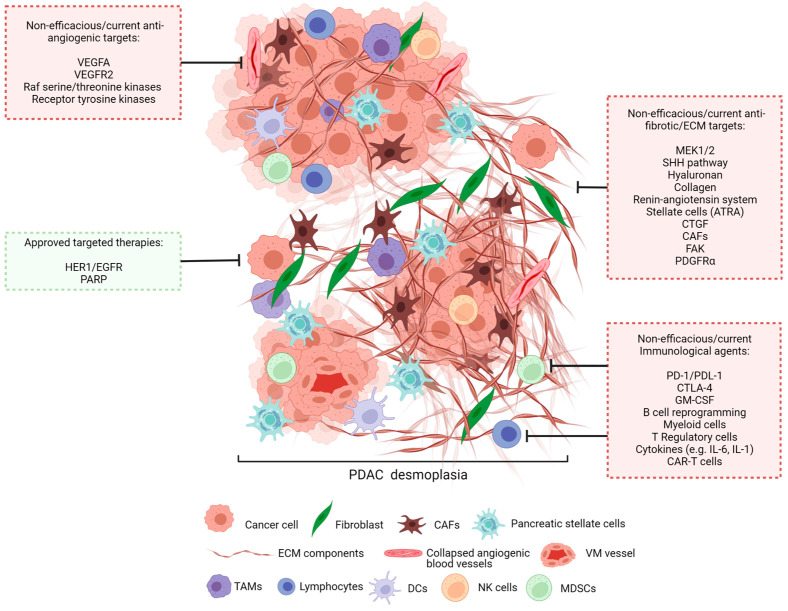
PDAC tumour microenvironment and therapeutic targets. A key feature of PDAC is the presence of dense fibrotic stroma or desmoplasia. PDAC tumours consist of ECM components such as collagen and hyaluronan, as well as pro-tumourigenic and anti-tumorigenic cellular components such as fibroblasts, vessel forming endothelial cells, vasculogenic mimicry (VM) vessels, cancer associated fibroblasts (CAFs), pancreatic stellate cells, lymphocytes, dendritic cells (DCs), tumour associated macrophages (TAMs), natural killer (NK) cells, and myeloid derived suppressor cells (MDSCs).

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
