# Peer review of "Overcoming the Fibrotic Fortress in Pancreatic Ductal Adenocarcinoma: Challenges and Opportunities"

_cancers, 2023, doi:10.3390/cancers15082354_

Round 1
Reviewer 1 Report
The review titled "Overcoming the fibrotic fortress in pancreatic ductal adenocarcinoma: challenges and opportunities" provides a comprehensive overview of the role of desmoplasia, the highly fibrotic and immunosuppressive TME, in the development and progression of PDAC. The authors discuss the challenges in delivering therapeutic drugs into the PDAC tumors and the need for continued/renewed efforts to improve the efficacy of therapy for better patient outcomes. The example of immunotherapy not showing significant improvement in PDAC patients compared to other cancers is a strong statement that highlights the urgent need for better treatments for PDAC.
Overall, the review is well-written and offers a detailed understanding of the complexity of PDAC and its TME. However, including another figure or diagram illustrating the most common targeted drugs (as shown in Tables) that relate to the cellular components of the PDAC TME (such as in Figure 2) would be helpful in enhancing readers' understanding of the topic.
Author Response
We thank the reviewer for the comment to add the most common targeted drugs (as shown in Tables) that relate to the cellular components of the PDAC TME. We have now altered Figure 2 on page 5, as below;

Reviewer 2 Report
In the present work, Min et al. reviewed the literature, with respect to the development of pancreatic ductal adenocarcinoma (PDAC), including the tumorigenic drivers that initiate and/or sustain the progression of the disease, as well as the complex and interwoven nature of the cellular and acellular components that come together to make PDAC one of the most aggressive and difficult to treat cancers. The authors also reviewed the known challenges in delivering drugs to PDAC tumours in concentrations that are therapeutic due to the presence of a highly fibrotic and immunosuppressive tumor microenvironment.
Their work is interesting and their manuscript is well-written.
The present work has merit for publication after some minor issues.
the authors should highlight their review findings as well as present the limitations in drug delivery, not only from the perspective of the microenvironment but also from the perspective of tumor mechanics i.e. why is pancreatic cancer so difficult to treat? is the severity due to the organ's significance, or due to the particularities of the tumor site?
Author Response
We appreciate the opportunity to clarify our review. We have now included additional text in the manuscript outlining the reasons why pancreatic cancer is difficult to treat on page 10, as below;
In addition to the late diagnosis of PDAC patients, there are also intrinsic and extrinsic factors that can impede upon therapy success. Such examples include acquired chemotherapy resistance, high mutational burden and pro-oncogenic signalling potential by the cancer cells, high metastatic burden, and the presence of an immunosuppressive TME. Furthermore, the dense stromal and connective tissue in PDAC tumours contributes to elevated tissue pressure through eliciting solid stress by compressing on the blood ves-sels, further inhibiting effective penetration of anti-cancer drugs, and thus contributing to a lack of efficiency for PDAC treatments.
Reviewer 3 Report
I read with great pleasure the paper by Kay K. Myo Min and colleagues, who reviewed the wealth of knowledge regarding the role of overabundant desmoplasia in the tumor microenvironment (TME) of the pancreatic ductal adenocarcinoma (PDAC), which creates a highly fibrotic and immunosuppressive TME that plays a crucial role in tumor development, progression, metastasis, and treatment resistance, especially for delivering drugs into the fortress of PDAC tumors.
Overall, the paper is interesting, nicely written, and well-arranged. The abstract is appropriate and to the point. The introduction provides a sufficient background with relevant current literature on the topic. I really appreciated the clarity of the tables and figures. The only minor point is that the Authors should state in the figure legends if the figures are original artworks from the Authors or are made on pre-made icons and templates from web applications, such as BioRender.
In my opinion, this paper may be worthy of publication in Cancers.
Author Response
We thank the reviewer for the comment. We generated the figures ourselves using BioRender using a paid subscription and have now included an acknowledgement to BioRender.com on page 14, as follows; Acknowledgements: Figures created using BioRender.com.